# Adverse health outcomes in offspring of parents with alcohol-related liver disease: Nationwide Danish cohort study

Peter Jepsen[1,2], Joe West[3,4,5], Anna Kirstine Kjær Larsen[6], Anna Emilie Kann[1,6,7], Frederik Kraglund[1], Joanne R. Morling[3,4], Colin Crooks[3,8], Gro Askgaard[1,6,7]*

1 Department of Hepatology and Gastroenterology, Aarhus University Hospital, Aarhus, Denmark, 2 Department of Clinical Epidemiology, Aarhus University Hospital, Aarhus, Denmark, 3 Nottingham University Hospitals NHS Trust and the University of Nottingham, NIHR Nottingham Biomedical Research Centre (BRC), Nottingham, United Kingdom, 4 Lifespan and Population Health, School of Medicine, University of Nottingham, Nottingham, United Kingdom, 5 Department of Clinical Medicine, Aarhus University, Aarhus, Denmark, 6 Center for Clinical Research and Prevention, Bispebjerg and Frederiksberg Hospital, Copenhagen, Denmark, 7 Section of Gastroenterology and Hepatology, Department of Medicine, Zealand University Hospital, Køge, Denmark, 8 Translational Medical Sciences, School of Medicine, University of Nottingham, Nottingham, United Kingdom

* gask@dadlnet.dk

## Abstract

### Background

Parental drinking can cause harm to the offspring. A parent's diagnosis of alcohol-related liver disease (ALD) might be an opportunity to reach offspring with preventive interventions. We investigated offspring risk of adverse health outcomes throughout life, their association with their parent's educational level and diagnosis of ALD.

### Methods and findings

We used nationwide health registries to identify offspring of parents diagnosed with ALD in Denmark 1996 to 2018 and age- and sex-matched comparators (20:1). We estimated the incidence rate ratios (IRRs) of hospital contacts with adverse health outcomes, overall and in socioeconomic strata. We used a self-controlled design to examine whether health outcomes were more likely to occur during the first year after the parent's ALD diagnosis. The 60,804 offspring of parents with ALD had a higher incidence rate of hospital contacts from age 15 to 60 years for psychiatric disease, poisoning, fracture or injury, alcohol-specific diagnoses, other substance abuse, and of death than comparators. Associations were stronger for offspring with low compared to high socioeconomic position: The IRR for admission due to poisoning was 2.2 versus 1.0 for offspring of an ALD parent with a primary level versus a highly educated ALD parent. Offspring had an increased risk for admission with psychiatric disease and poisoning in the year after their parent's ALD diagnosis. For example, among offspring whose first hospital contact with psychiatric disease was at age 13 to 25 years, the IRR in the first year after their parent's ALD diagnosis versus at another time was 1.29 (95% CI 1.13, 1.47). Main limitation was inability to include adverse health outcomes not involving hospital contact.

**Data Availability Statement:** Electronic health records are, by definition, considered "sensitive" data in Denmark by the Data Protection Act and

cannot be shared via public deposition because of information governance restrictions in place to protect patient confidentiality. In accordance with Danish law, we have only been allowed to work with the data on a Trusted Research Environment (in Danish: Forskermaskinen). We are not permitted to download the data, or to share it. However, others can request the same data that we used, and this can be done via an application to Statistics Denmark emailed to forskningsservice@dst.dk. More information on the application procedure can be found at (https://www.dst.dk/en/TilSalg/Forskningsservice/Dataadgang). The code that we used to conduct the analyses is available upon reasonable request, however, we can only share the code with others who work within the Secure Research Environment of Statistics Denmark. Registry-based studies does not require ethical approval in Denmark.

**Funding:** The study was funded by the Novo Nordisk Foundation (NNF: NNF18OC0054612, https://novonordiskfonden.dk/) donated to PJ and the 'Savværksejer Jeppe Juhl og hustru Ovita Juhls Mindelegat' foundation, also donated to PJ (https://www.legatbogen.dk/savvrksejer-jeppe-juhls-og-hustruovita-juhls-mindelegat-med-flere/stoetteomraade/10229). The funders had no role in study design, data collection and analysis, decision to publish, or preparation of the manuscript.

**Competing interests:** The authors have declared that no competing interests exist.

**Abbreviations:** ALD, alcohol-related liver disease; ICD, International classification of disease; IR, incidence rate; IRD, incidence rate difference; IRR, incidence rate ratio; ISCED, International Standard Classification of Education; PY, person years; SCCS, self-controlled case series.

## Conclusions

Offspring of parents with ALD had a long-lasting higher rate of health outcomes associated with poor mental health and self-harm that increased shortly after their parent's diagnosis of ALD. Offspring of parents of low educational level were particularly vulnerable. This study highlights an opportunity to reach out to offspring in connection with their parent's hospitalization with ALD.

## Author summary

### Why was this study done?

- Young offspring of parents with alcohol use disorder have higher rates of poor mental health, and substance misuse and many of them have not received professional help in relation to their parent's alcohol misuse.

- It has not been investigated whether these higher rates of adverse health outcomes continue throughout adulthood are influenced by their parent's diagnosis of alcohol-related liver disease (ALD) and the influence of parental socioeconomic position has not been thoroughly investigated.

- The temporal relationship between a parent's diagnosis of ALD and an offspring's adverse health outcomes may show us an opportune time to offer support to the offspring in the clinical setting, hopefully preventing future adverse outcomes.

### What did the researchers do and find?

- This was an exploratory study of adverse health outcomes in 60,804 offspring of patients with ALD compared to 1.2 million matched controls, based on Danish nationwide health registries 1996 to 2018.

- Offspring of 15 to 60 years had higher rates of hospital contacts due to psychiatric disease, poisoning, fracture or injury, alcohol-specific diagnoses, other substance abuse, and of death than comparators had. Offspring of parents of low compared to higher educational level had the highest rates of hospital contacts.

- In the year after the parent's diagnosis of ALD, offspring experienced an increase in hospitalization due to outcomes associated with self-harm (e.g., poisoning) and poor mental health (e.g., psychiatric disease).

### What do these findings mean?

- Preventive interventions are needed to help offspring of parents with alcohol use disorder, and a parent's hospital admission for ALD is a window of opportunity for healthcare professionals to reach out.

- The main study limitation was inability to include adverse health outcomes not involving hospital contacts leading to an underestimated burden of adverse health outcomes in offspring of parents with ALD.

## Introduction

Hazardous alcohol consumption can cause harm not only to the individual who drinks alcohol but also to their family, including offspring [1]. Worldwide, around 100 million people may be affected by the substance use of a close relative [2]. Offspring of parents with alcohol use disorder are more likely to experience family dysfunction, material deprivation, domestic violence, and childhood abuse [1,3–5]. It is therefore not surprising that young offspring of parents with alcohol use disorder have higher rates of outcomes associated with stressful life circumstances; poor mental health; and self-harm, including depression, injuries, fractures, heavy alcohol drinking, and drug use, according to a systematic review from 2016 [1,6–8]. However, the literature has not covered whether these risks continue throughout adulthood, except for an increased risk of somatic disease in adulthood, reported in more recent studies [9,10]. In addition, it has not been covered whether parental socioeconomic position influence the risk of outcomes associated with self-harm such as poisoning and injury in offspring of parents with alcohol use disorder [11].

Offspring of parents with alcohol-related liver disease (ALD) may have been exposed to their parent's alcohol misuse for 20 years since this is the average of time of heavy drinking that precede a diagnosis of ALD [12]. Such offspring may witness their parent's acute hospital admissions with impaired consciousness due to liver coma, abdominal swelling, and vomiting blood due to variceal bleeding and even death as 25% of patients diagnosed with ALD die within the first year [12,13]. Parental ALD may therefore exaggerate mental stress in the offspring which could be observed as an increased risk of hospital admissions with psychiatric disease or intentional poisoning for the offspring at this time point [8,14].

Offspring of parents with alcohol use disorder may not receive the help they need: It is estimated that 40% to 90% of offspring have not received professional help in relation to their parent's alcohol misuse [15–17]. According to a recent systematic review, this may be due to the circumstance that the offer of professional help to the offspring often depends on their parent seeking treatment for alcohol use disorder, but the parent may not do so [18,19]. Therefore, the clinical setting has been suggested as another arena to offer preventive care to the offspring of parents with alcohol use disorder [20]. The temporal relationship between a parent's diagnosis of ALD and an offspring's risk of hospitalization due to psychiatric disease or poisoning may show us an opportune time to offer support to the offspring in the clinical setting, hopefully preventing future adverse outcomes.

Given this background, we conducted an exploratory study of incidence rates of hospital contacts with diagnoses associated with poor mental health, self-harm, accidents, violence, and substance abuse in offspring of parents with ALD compared to matched comparators [14]. Our aim was to describe these risks and not to address the causes behind them and we did not have prespecified hypotheses to test. Specifically, we estimated incidence rate ratios (IRRs) of hospital contacts according to psychiatric disease, poisoning, fracture or injury, alcohol-specific diagnoses, other substance abuse, and death in offspring of parents with ALD and matched comparators. We examined these associations overall and within subgroups defined

by sex and socioeconomic position and across time, age, and time since the parent's diagnosis of ALD.

## Methods

The study was based on Danish healthcare registries that allow accurate parent–child linkage [21]. We identified all offspring of patients diagnosed with ALD in 1996 to 2018, and comparators matched to those offspring on age, sex, and birth year. We followed offspring and comparators until 1 January 2019 for diagnoses of adverse health outcomes and estimated their relative risk of these outcomes.

### Data sources

We used data from 4 Danish nationwide registries: The National Patient Registry [22], the Registry of Cause of Death [23], the Civil Registration System [21], and the Population Education Registry [24]. All the included data sources are described in more detail in S1 Table.

Virtually all healthcare in Denmark is provided by the national health authorities, allowing true population-based register-linkage studies covering all inhabitants of Denmark. Data were linked by use of the personal identification number, a unique identifier assigned to all Danish inhabitants since 1968 [21]. All linkages and identification of comparators were performed within Statistics Denmark, a governmental institution that collects and processes information for various administrative and scientific purposes. We accessed and analyzed the data using Statistics Denmark's secure researcher environment.

### Offspring of parents with alcohol-related liver disease

We included all offspring who were alive when their parent was diagnosed with ALD during 1996 to 2018 excluding offspring born after parental diagnosis of ALD. We first identified all patients given a diagnosis with ALD (ICD-10: K70.x) in the hospital or as a cause of death in the National Patient Registry and the Cause of Death registry during the 1996 to 2018 period [22,23]. We restricted the study period to when diagnoses were coded in accordance with the International Classification of Diseases, edition 10 (ICD-10), which began in 1994, to ensure homogeneity of coding practices [22]. We then used the parent–child linkage in the Civil Registration System to identify offspring of all these patients diagnosed with ALD in the hospital or as a cause of death [21]. The misclassification of biological parent–child relations in the Danish Civil Registration System is 1% and, therefore, negligible. We included all offspring born any time before their parent was diagnosed with ALD in the study, but we did not include offspring born after their parent's diagnosis of ALD ($n$ = 910).

We used the educational attainment of the parents with ALD as a proxy for socioeconomic status in the offspring. Educational attainment that occurred prior to adulthood may be less influenced by harmful alcohol consumption during adulthood in the parent with ALD than are other proxies of socioeconomic position such as employment status and income [25]. It was a limitation that we did not have data on the educational attainment of the offspring. However, an offspring's final educational attainment may not have been reached at the time of the parent's ALD diagnosis [26]. Information on the highest educational attainment came from the Population Education Registry [24]. About 3% of the population have unknown educational status, either because they are immigrants to Denmark or because their education is not acknowledged by Danish authorities. We grouped educational attainment according to the International Standard Classification of Education (ISCED), noting that Denmark has no educational program that corresponds to ISCED level 4, post-secondary non-tertiary education [27]. Briefly, we categorized the highest attained education as "primary level" equivalent to

≤10 years of education; "secondary level" equivalent to 11–13 years of education; "higher": ≥14 years of education. S1 Fig provides a detailed description of our education categorization in the study.

## Comparators

We identified 20 comparators for each offspring of a patient diagnosed with ALD among the general Danish population. Matching took place on the date of the parent's first-time ALD diagnosis, and comparators had to be alive and without a diagnosis of ALD on that date. Comparators also had to be born in the same year and have the same sex as offspring of patients with ALD to whom they were matched. We chose to match on birth year in addition to age and sex, since the pattern of hospital care has changed during the study period with an increasing frequency in hospital contacts [28]. They were otherwise selected at random from the general Danish population, with replacement, meaning that a comparator could be selected as a comparator for multiple offspring. This method ensures that each control is a truly randomly sampled person from the full population, not conditional on the previous cases' sampling [29].

## Outcomes

We followed the offspring of patients with ALD and their matched comparators from the offspring's birthday or 1 January 1996 (whichever was the latest) until death, emigration, or 1 January 2019 (whichever came first). This means that offspring born after 1 January 1996 were followed from birth, while offspring born before 1 January 1996 were followed from 1 January 1996. For every hospital contact, the treating physician specifies 1 primary diagnosis and up to 20 secondary diagnoses. During the observation period, offspring and comparators were followed for these hospital contacts with a primary or secondary diagnosis that could occur repeatedly: psychiatric diseases other than substance abuse (ICD.10: F20.x–F99.x), intentional or accidental poisoning (emergency room visits and acute inpatient admissions only, ICD-10: T36.x–T62.x, T64.x, T65.x, X60.x–X84.x), fracture or injury (emergency room visits only, ICD-10: Sxx.x, T0x.x–T35.x, T66.x–T98.x), alcohol-specific diagnoses (ICD-10: K70.x, F10.x, E24.4, E52.x, G31.2, G62.1, G72.1, I42.6, K29.2, K85.2, K86.0, O35.4, P04.3, Q86.0, R78.0, T51.0, T51.1, T51.9, Y15.x, Y90.x, Y91.x) [30], abuse of other substances than alcohol (ICD-10: F11.x–F19.x), or all-cause death. However, in a sensitivity analysis, we restricted outcomes to hospital contacts with only a primary diagnosis code for an adverse health outcome. S2 Table provides the most frequently occurring hospital diagnoses for offspring and comparators during follow-up.

## Statistical analysis

We computed incidence rates of the adverse health outcomes for offspring and comparators, including incidence rate ratios and incidence rate differences. Analyses were conducted overall and within subgroups according to sex and the educational attainment of the parent with ALD.

We examined variation in IRR over time for offspring versus comparators and we used 3 different time scales in these analyses: offspring age, calendar time, and time since the parent's diagnosis of ALD.

**Offspring age and calendar time.** We computed the IRR of adverse health outcomes from age 18 years to age 19 years for those who were under observation at age 18 years (and similarly, we computed the IRR from age 17 to age 18, from age 19 to age 20, from age 20 to age 21 years, etc.). We also computed the IRR from 1 January 2004 to 31 December 2004, from 1 January 2005 to 31 December 2005, etc.

**Time since the parent's diagnosis of alcohol-related liver disease.** We computed the IRR from 1 year before the index date to the index date, from the index date to 1 year after the index date, etc. In addition, we used a self-controlled case series (SCCS) method to eliminate confounding due to time-constant characteristics of the offspring and thus strengthen the evidence in favor of a causal effect of the parent's diagnosis of ALD on the offspring's adverse health outcomes. Briefly, the SCCS method investigates the temporal association between a transient exposure (diagnosis of ALD in the parent) and an event (adverse health outcome in the offspring) [31]. Comparisons in the SCCS method are made only within individuals who have experienced an event whereby individuals act as their own control.

Applied in this study, the SCCS analysis examined whether, among offspring hospitalized for an adverse outcome within a specific age range, their first hospitalization for that outcome was more likely to occur during the first year after their parent was diagnosed with ALD diagnosis compared to at any other time within that age range [31]. We conducted these analyses within 3 age ranges: one for offspring who had their first hospitalization for an adverse health outcome at age 0 to 12 years, one for age 13 to 25 years, and one for age 26 to 60 years. We used the R package SCCS for these analyses [32,33].

Study reporting guideline: This study is reported as per the Strengthening the Reporting of Observational Studies in Epidemiology (STROBE) guideline (S3 Table). Our prespecified analysis plan was the data application of the study sent to Statistics Denmark (written in Danish) that can be shared with interested readers upon request to the corresponding author. This plan did not include definition of exact outcomes or exact methods.

## Patient and public involvement

We did not include patients and the public directly throughout the research process (formulation of research questions, outcome measures development, study design, recruitment, the conduct of the study, and dissemination of the results).

## Results

We included 60,804 offspring of patients with a diagnosis of ALD from 1996 to 2018 in Denmark (Table 1). The offspring had a median age of 31.8 years (IQR 23.4 to 39.4) when their parent was diagnosed with ALD, with 5,087 (8.4%) being younger than 15 years and 12,695 (21%) being 15 to 24 years. Most commonly, it was the father [$n = 40,073$ (65.9%)] who was diagnosed with ALD, and in 944 (1.6%) offspring both parents were diagnosed with ALD. The 60,804 offspring were born to 30,061 different parents who were subsequently diagnosed with ALD. Of those 30,061 parents, 9,063 (30%) had 1 offspring included in our analysis, 13,270 (44%) had 2 offspring included, 5,538 (18%) had 3 offspring included, and 2,190 (7.3%) had 4 or more offspring included. There were 1,216,047 matched comparators, and 1,015,399 (83.5%) of them were unique individuals. S2 Fig shows the number of offspring and comparators under follow-up according to current age.

Offspring had a higher incidence rate of hospital contacts with psychiatric disease, poisoning, fracture or injury, an alcohol-specific diagnosis, abuse of other substances than alcohol, and they also had a higher mortality rate than comparators (Table 2). Hospital contacts due to poisoning were most frequently due to poisoning with medications that are intentional in their nature, whereas hospital contacts with psychiatric disease were most frequently due to stress, depression, or anxiety (S2 Table). The highest IRR was for alcohol-specific diagnoses (IRR = 2.29, 95% CI: 2.26, 2.32), $p < 0.001$, and the lowest was for hospital contacts with fracture or injury (IRR = 1.23, 95% CI: 1.22, 1.23), $p < 0.001$. However, since fracture or injury was a frequent outcome, it resulted in the largest number of extra

**Table 1. Baseline characteristics of offspring of patients with alcohol-related liver disease in Denmark 1996–2018 and their matched comparators.**

|  | Offspring | Comparators |
|---|---|---|
| Number | 60,804 | 1,216,047 |
| Sex, % men | 51.5% | 51.5% |
| **Age at the time of parents' ALD diagnosis** |  |  |
| Median (IQR) | 31.8 (23.4–39.4) | 31.8 (23.4–39.4) |
| <15 years | 5,087 (8.4%) | 101,738 (8.4%) |
| 15–24 years | 12,695 (20.9%) | 253,893 (20.9%) |
| 25–34 years | 19,168 (31.5%) | 383,346 (31.5%) |
| 35–44 years | 17,265 (28.4%) | 345,293 (28.4%) |
| 45–54 years | 5,981 (9.8%) | 119,618 (9.8%) |
| 55–64 years | 601 (1.0%) | 12,019 (1.0%) |
| ≥65 years | 7 (0.01%) | 140 (0.01%) |
| **Calendar year of birth** |  |  |
| Before 1960 | 3,785 (6.2%) | 75,695 (6.2%) |
| 1960–1969 | 15,654 (25.8%) | 313,077 (25.8%) |
| 1970–1979 | 19,236 (31.6%) | 384,709 (31.6%) |
| 1980– | 22,129 (36.4%) | 442,566 (36.4%) |
| **Parent with ALD** |  |  |
| Father has ALD | 40,073 (65.9%) | - |
| Mother has ALD | 19,787 (32.5%) | - |
| Both parents have ALD | 944 (1.6%) | - |
| **Socioeconomic position of ALD parent** |  |  |
| ALD parent has primary level of education | 24,155 (39.7%) |  |
| ALD parent has secondary level of education | 25,999 (42.8%) |  |
| ALD parent has higher level of education | 8,679 (14.3%) |  |
| ALD parent has an unknown education level | 1,971 (3.2%) |  |

ALD, alcohol-related liver disease; IQR, interquartile range.

hospital contacts for offspring [26.7 (95% CI: 26.1, 27.4) extra hospital contacts per 1,000 person years (PY), $p < 0.001$] (Table 2). IRR did not change when only hospital contacts with primary and not secondary diagnoses codes counted as a hospital contact for psychiatric disease, poisoning, fracture or injury, an alcohol-specific diagnosis, or abuse of other substances than alcohol (S4 Table).

**Table 2. Incidence rates, incidence rate ratios, and incidence rate differences for adverse health outcome in offspring ($n = 60,804$) of parents with alcohol-related liver disease in Denmark 1996–2018 and their matched comparators ($n = 1,213,356$).**

|  | Events among offspring | IR per 1,000 PY among offspring | IRR vs. comparators | IRD vs. comparators per 1,000 PY |
|---|---|---|---|---|
| **Hospital diagnosis** |  |  |  |  |
| Psychiatric disease | 54,495 | 39.7 (39.3–40.0) | 1.45 (1.44–1.46) | 12.3 (12.0–12.7) |
| Intentional or accidental poisoning | 7,437 | 5.4 (5.3–5.5) | 1.74 (1.70–1.78) | 2.3 (2.2–2.4) |
| Fracture or injury | 197,085 | 143.5 (142.9–144.1) | 1.23 (1.22–1.23) | 26.7 (26.1–27.4) |
| Alcohol-specific | 22,388 | 16.3 (16.1–16.5) | 2.29 (2.26–2.32) | 9.2 (9.0–9.4) |
| Other abuse | 11,364 | 8.3 (8.1–8.4) | 2.13 (2.09–2.17) | 4.4 (4.2–4.5) |
| Death | 1,448 | 2.1 (2.0–2.2) | 1.53 (1.45–1.62) | 0.7 (0.6–0.8) |

Subgroup analyses showed higher IRR in the offspring of mothers than in the offspring of fathers with ALD concerning hospital contacts due to psychiatric disease, an alcohol-specific diagnosis, and abuse of other substances than alcohol. IRRs for adverse health outcomes were higher for male than for female offspring, except for hospital contacts due to fracture or injuries and other abuse than alcohol, for which no difference according to sex was found. IRRs were highest for offspring with both parents diagnosed with ALD (Fig 1). There was evidence of socioeconomic inequality, with IRR of all adverse health outcomes being highest for offspring whose parent with ALD had the lowest educational attainment. For example, the IRR for an alcohol-specific hospital contact was 2.62 (95% CI: 2.57, 2.67), $p < 0.001$, for offspring of an ALD parent with a primary-level education compared with 1.61 (95% CI: 1.54, 1.68), $p < 0.001$, for offspring of a parent with a higher-level education.

### Influence of offspring age and calendar time

The IRR of all adverse outcomes for offspring versus comparators peaked during adolescence or young adulthood and declined thereafter, except for hospital contacts with alcohol-specific diagnoses that remained high during adulthood and declined after the age of 60 years (Fig 2). Analyses of IRR by calendar time showed that IRR remained stable during the 1996 to 2018 period (S3 Fig).

### Influence of time since the parent's diagnosis of alcohol-related liver disease

For most of the outcomes, the IRR was stable from 5 years before the parent's ALD diagnosis to 10 years after (Fig 3). On the contrary, there was a sharp increase in the IRR for hospitalization with poisoning in the year after the parent's diagnosis of ALD. An increase in the IRR was also seen for hospitalization with psychiatric disease and for death in the year after the parent's diagnosis of ALD, but not for the other adverse health outcomes.

The self-controlled analyses showed that offspring were generally more likely to experience their first adverse health outcome during the year following their parent's diagnosis of ALD than they were at another time (Table 3). This pattern was most pronounced for those whose first such outcome was at age 26 to 60 years, but it was also seen for those whose debut with an adverse outcome occurred at a younger age. Specifically, among offspring whose first admission for poisoning was between age 13 and 25 years, that first admission was 1.25-fold more likely (95% CI 1.01, 1.55), $p = 0.04$, to occur during the year after a parent was diagnosed with ALD than at any other time between age 13 and 25 years. The same phenomenon was seen for psychiatric disease, IRR = 1.29 (95% CI 1.13, 1.47), $p < 0.001$.

### Discussion

This study of 60,000 offspring of a parent with ALD showed that offspring had higher rates of hospital contacts from the age of 15 to 60 years for a range of adverse health outcomes including psychiatric disease, poisoning, fracture or injury, alcohol-specific diagnoses, other substance abuse, and of death than comparators had. Offspring of parents of low compared to higher educational level were particularly vulnerable. Finally, offspring experienced a sharp increase in the rate of hospitalization due to intentional poisoning and an increase in hospitalization with psychiatric disease in the year after their parent's diagnosis of ALD compared to controls. The self-controlled analyses confirmed that offspring were generally more likely to experience their first adverse health outcome during the year following their parent's diagnosis of ALD than they were at another time.

This study is the first to show the extent of the life-long burden on these offspring's health. In addition, we have identified a surge in outcomes associated with poor mental health and

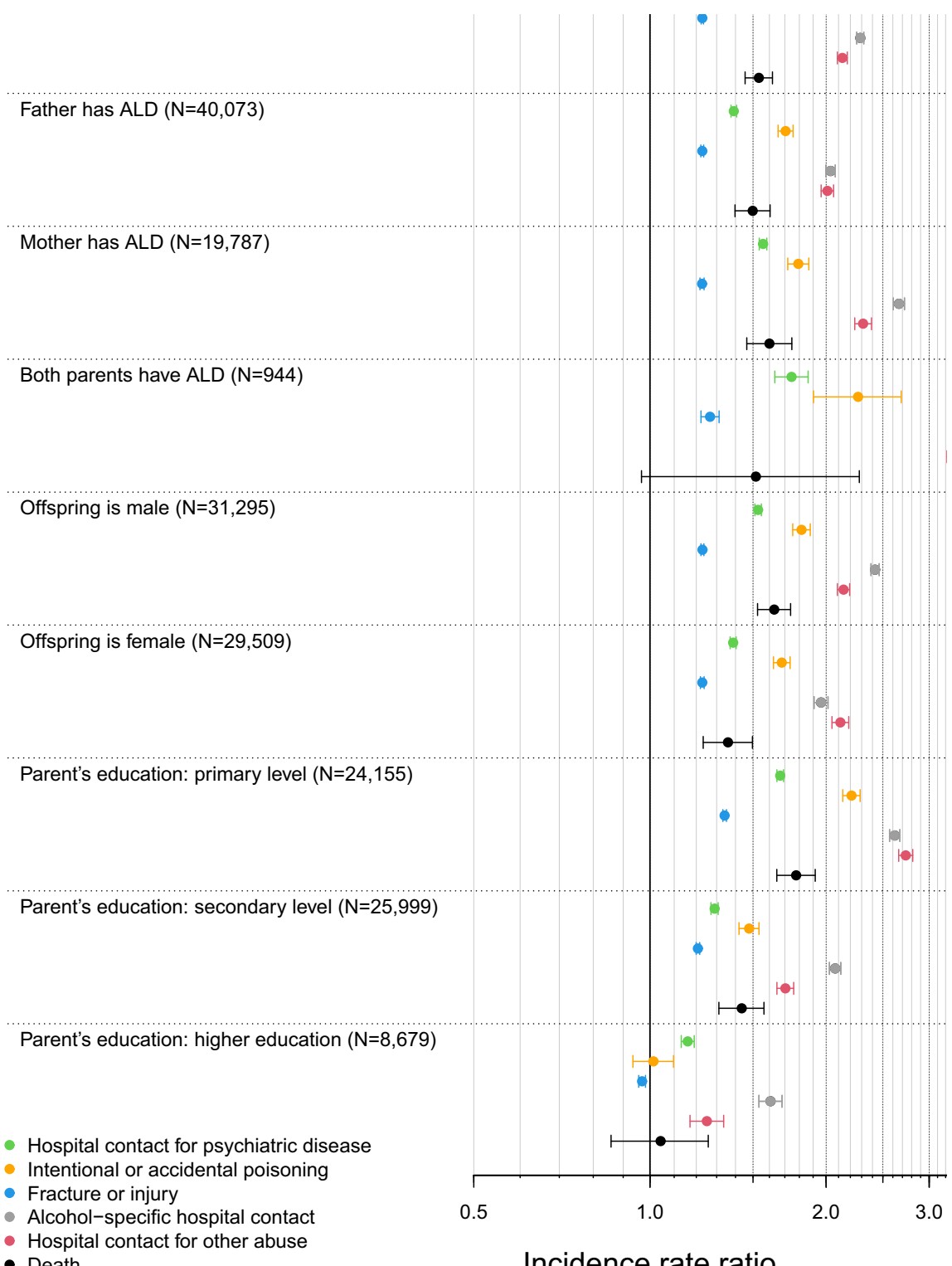

**Fig 1. Forest plot showing incidence rate ratios of adverse health outcome in offspring of patients (*n* = 60,804) with alcohol-related liver disease in Denmark 1996–2018 versus matched comparators (*n* = 1,216,047), overall and in subgroups.**

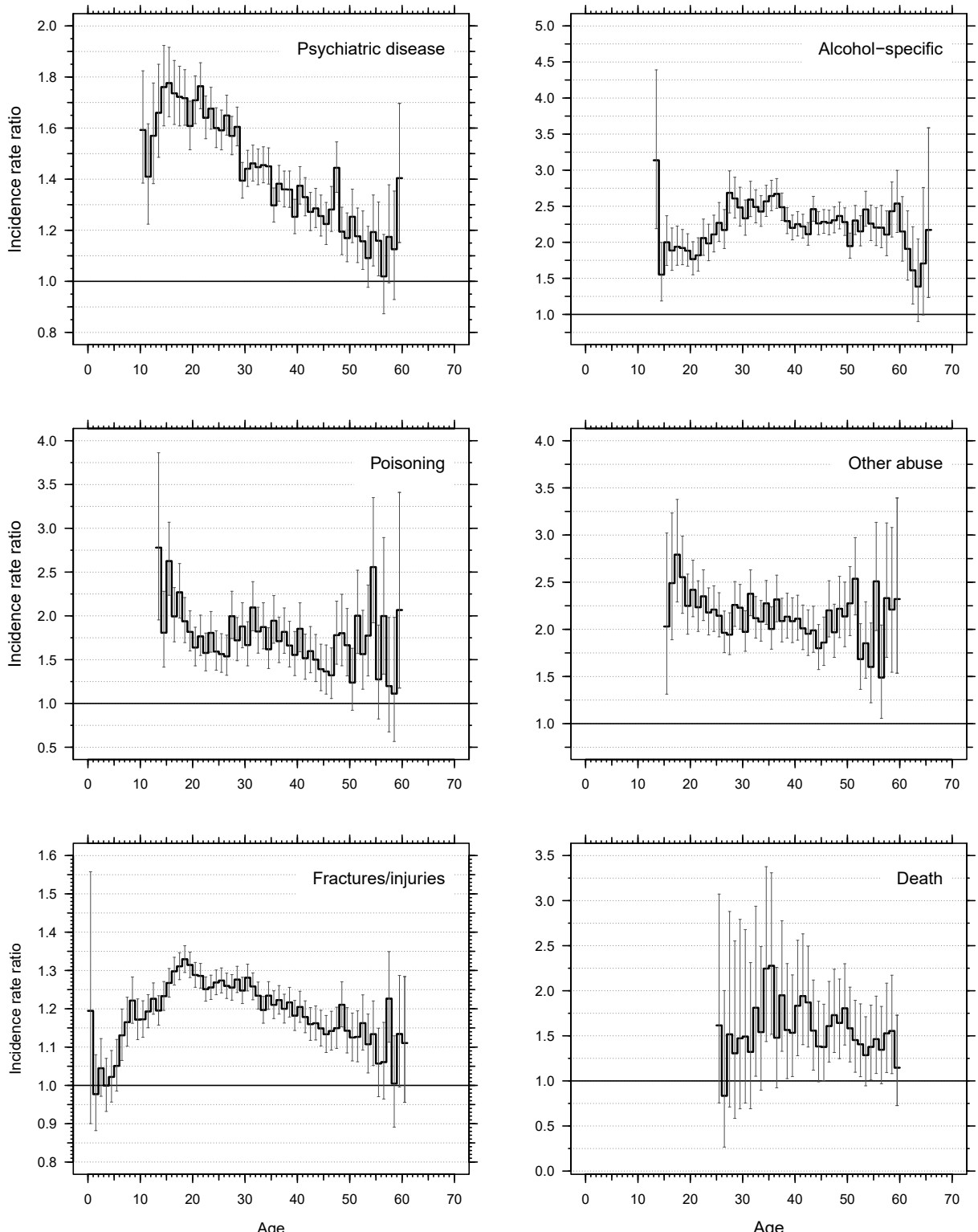

**Fig 2. Age-specific incidence rate ratios of hospital contacts with adverse health outcomes for offspring vs. matched comparators.** Age-specific incidence rates are computed separately for offspring and comparators, and the ratio between them (the incidence rate ratio) is shown here with short horizontal lines. The vertical bars illustrate 95% confidence intervals.

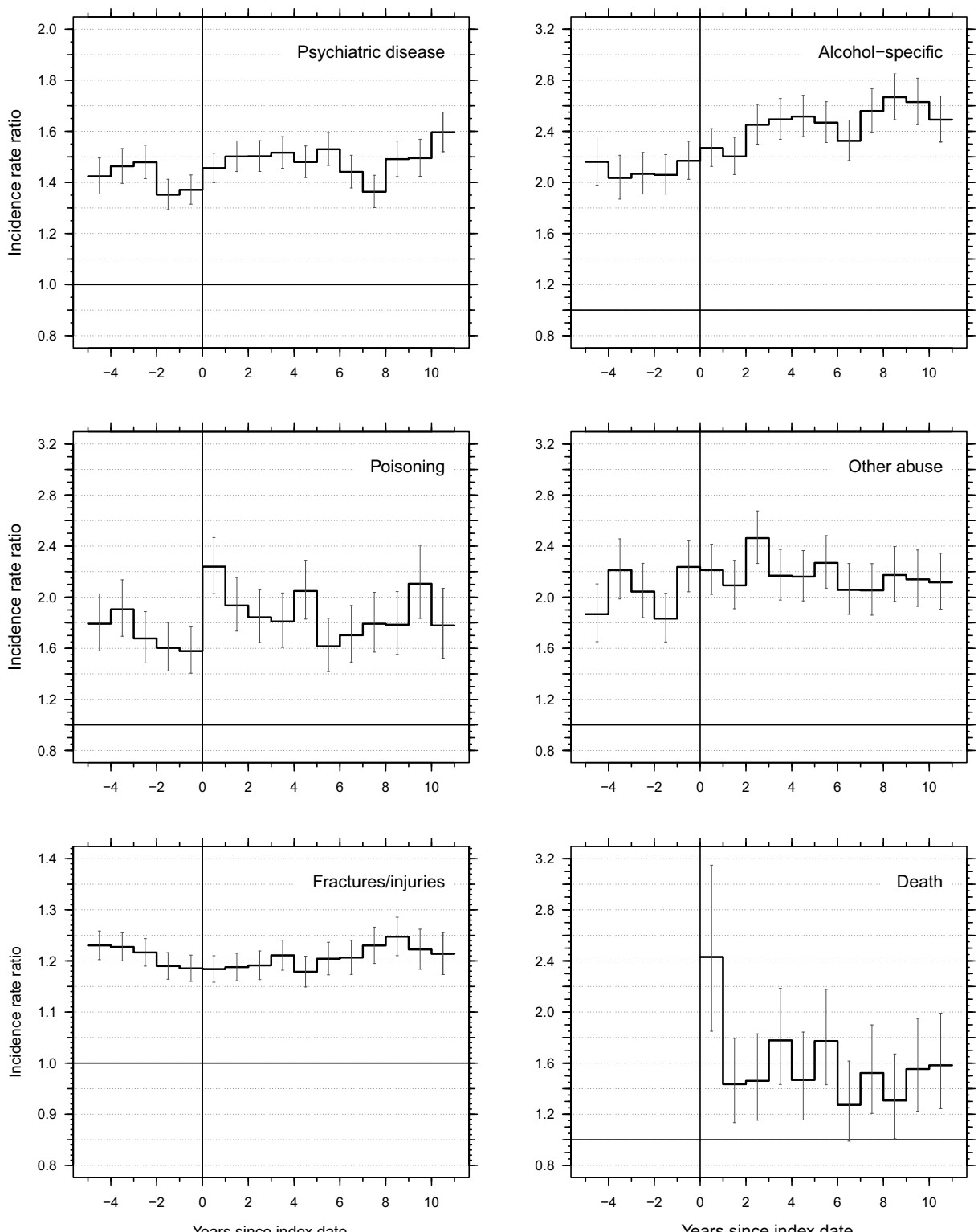

**Fig 3. Incidence rate ratios of hospital contacts with adverse health outcome in offspring of patients (*n* = 60,804 with alcohol-related liver disease in Denmark 1996–2018 versus matched comparators (*n* = 1,216,047) according to time since parents' diagnosis of alcohol-related liver disease.** The vertical capped bars illustrate 95% confidence intervals, while the vertical line indicates the time of the parent's diagnosis of alcohol-related liver disease.

**Table 3. Self-controlled cases series examining whether a first hospitalization for an adverse health outcome (or death) is more likely to occur in the year after a parent is diagnosed with alcohol-related liver disease.**

| Hospital diagnosis | Incidence rate ratio for hospitalization due to an adverse health outcome during the first year after parent's diagnosis of alcohol-related liver disease vs. any other time among offspring whose first hospitalization for the adverse health outcome was at . . . | | |
|---|---|---|---|
| | Age 0–12 years | Age 13–25 years | Age 26–60 years |
| Psychiatric disease | 1.14 (0.78–1.68) | 1.29 (1.13–1.47) | 1.22 (1.10–1.36) |
| Intentional or accidental poisoning | 2.14 (0.93–4.93) | 1.25 (1.01–1.55) | 1.42 (1.19–1.69) |
| Fracture or injury | 1.12 (0.97–1.29) | 1.10 (1.01–1.18) | 1.05 (0.99–1.12) |
| Alcohol-specific | 1.16 (0.41–3.27) | 1.15 (0.93–1.42) | 1.33 (1.16–1.54) |
| Other abuse | N/A | 0.99 (0.75–1.30) | 1.23 (1.01–1.48) |

self-harm, that is hospitalization due to poisoning and psychiatric disease, shortly after a parent's diagnosis of ALD, highlighting an opportunity to reach out to these offspring with preventive interventions when their parent is seen at the hospital for ALD.

We have expanded our understanding of socioeconomic inequalities in alcohol-related harms brought upon offspring: Previous studies have shown that children of a family with low socioeconomic position are more vulnerable to the consequences of a family member's alcohol misuse in terms of low mental health, alcohol, and drug use; we have shown that it also applies to fractures and injuries, poisonings, and death [11].

The offspring we studied seem to be more vulnerable than children of parents with other chronic diseases. Our finding of a persistently increased risk of adverse health outcomes in offspring of parents with ALD contrasts with the short-lived (2 years) increase in the risk for suicidal thoughts and attempts in adolescent offspring after their parent's cancer diagnosis [34]. The rate of psychiatric hospital contacts that we observed in offspring of parents with ALD was about 4-fold higher than what was found for Danish children with a family member who died or was diagnosed with a life-threatening disease [20]. Several circumstances could explain why the offspring of parents with alcohol misuse are more vulnerable to adverse health outcomes than the offspring of parents with other chronic diseases: Offspring of parents with alcohol misuse are more likely to encounter psychosocial stressors, such as growing up in a broken family, experiencing domestic violence, being victims of abuse, and suffering material deprivation [1,3]. In addition, the time after the parent's diagnosis is revealed can be even more stressful to the offspring: A diagnosis of ALD is often preceded by many years of continued heavy drinking, and a frequent dramatic climax of this ordeal is that the parent vomits blood and/or goes into hepatic coma, is hospitalized and diagnosed with ALD, and then dies less than a year later [12,13].

The Danish healthcare system and nationwide healthcare databases are well-suited for a study like ours. We could identify a nationwide, large cohort of offspring of patients with ALD with complete coverage and follow-up, thus avoiding the bias from non-participation by vulnerable groups and the bias from self-reporting in health surveys [35]. Our study uses the general population as the comparator, thereby minimizing the risk of selection bias for comparators. Our study has some limitations, too. First, we are likely to have underestimated the number of injuries or mental health disorders that did not result in a hospital contact. This is important when studying outcomes associated with self-harm (injuries, poisonings) or violence since only a fraction of these lead to hospital contacts (S2 Table) [14]. However, the underestimation in outcomes may not affect the relative difference in outcomes between offspring and comparators that we calculated as IRRs. Second, we did not know whether the

parent with ALD had a close relationship or lived with their offspring which was a limitation of the study. However, it is safe to assume that most offspring had a close relationship with their parent, as, in Denmark, the large majority of children have relations with both of their parents, even when parents are divorced [36]. Third, it was a limitation of our study that we did not match comparators and offspring on parental socioeconomic status. However, even offspring of a parent with ALD of higher education had increased risk of hospital contacts due to psychiatric disease, alcohol-specific, and other abuse compared to comparators of parents from a broader variety of educational level. Thus, our study findings could not be explained away by confounding by educational level of the parent with ALD.

Our self-controlled analysis was a strong point. In this analysis, every offspring served as his/her own control, minimizing confounding by offspring factors that remained unchanged through the study period. Thus, we are confident that a parent's diagnosis of ALD is associated with a causal role in the adverse outcomes for the offspring. Even so, the exact nature of this role remains uncertain, and there may be a multitude of other contributing causes. More research is needed to disentangle the many factors that drive the higher rates of adverse health outcomes. Importantly, results from our study highlight the need for preventive interventions and the design of such interventions may not necessarily depend on detailed knowledge of causal factors.

Today, most offspring of parents with alcohol use disorder have not received professional help in relation to their parents drinking [16,17]. A parent's hospital admission for ALD is a window of opportunity for healthcare professionals to reach out. Few interventions have been tested to improve outcomes in offspring of parents with alcohol misuse [18,19], and the associations we describe have not changed in this millennium (S3 Fig). We hope that this study will motivate further efforts to provide the help that these offspring obviously need.

In conclusion, this study showed a long-lasting higher rate of health outcomes associated with poor mental health, self-harm, and substance abuse in offspring of parents with ALD that increased shortly after their parent's diagnosis. Offspring of parents of low educational level were particularly vulnerable. This study highlights an opportunity to reach out to offspring in connection with their parent's hospitalization with ALD.

## Transparency statement

Author PJ and GA affirms that the manuscript is an honest, accurate, and transparent account of the study being reported, that no important aspects of the study have been omitted.

## Supporting information

**S1 Table. Included data sources.**
(DOCX)

**S2 Table. The 10 most frequent hospital contacts in offspring and comparators.**
(DOCX)

**S3 Table. STROBE Statement.**
(DOCX)

**S4 Table. Incidence rates, incidence rate ratios and differences for adverse health outcome in offspring of patients with alcohol-related liver disease in Denmark 1996–2018 and their matched comparators, based on primary diagnosis codes.**
(DOCX)

**S1 Fig. The Danish Education System.**
(DOCX)

**S2 Fig. Number of offspring of patients with ALD and their matched comparators under observation, by current age.**
(DOCX)

**S3 Fig. Calendar year-specific incidence rate ratios of adverse health outcomes for offspring vs. matched comparators.**
(DOCX)

## Author Contributions

**Conceptualization:** Peter Jepsen, Joe West, Colin Crooks, Gro Askgaard.

**Formal analysis:** Peter Jepsen.

**Funding acquisition:** Peter Jepsen.

**Investigation:** Peter Jepsen, Gro Askgaard.

**Methodology:** Peter Jepsen, Joe West, Anna Kirstine Kjær Larsen, Anna Emilie Kann, Frederik Kraglund, Joanne R. Morling, Colin Crooks, Gro Askgaard.

**Resources:** Peter Jepsen.

**Software:** Peter Jepsen.

**Supervision:** Peter Jepsen, Joe West, Colin Crooks.

**Writing – original draft:** Gro Askgaard.

**Writing – review & editing:** Peter Jepsen, Joe West, Anna Kirstine Kjær Larsen, Anna Emilie Kann, Frederik Kraglund, Joanne R. Morling, Colin Crooks.

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
