## [Editor Report · Decision Letter 0]

19 Apr 2024

Dear Dr Askgaard, 

Thank you for submitting your manuscript entitled "Adverse health outcomes in offspring of parents with alcohol-related liver disease: nationwide Danish cohort study" for consideration by PLOS Medicine.

Your manuscript has now been evaluated by the PLOS Medicine editorial staff as well as by an academic editor with relevant expertise and I am writing to let you know that we would like to send your submission out for external peer review.

Please re-submit your manuscript within two working days, i.e. by Apr 23 2024 11:59PM.

Kind regards,

Katrien G. Janin, PhD

Senior Editor

PLOS Medicine

---

## [Decision Letter · Decision Letter 1]

14 Jun 2024

Dear Dr. Askgaard,

Thank you very much for submitting your manuscript "Adverse health outcomes in offspring of parents with alcohol-related liver disease: nationwide Danish cohort study" (PMEDICINE-D-24-01249R1) for consideration at PLOS Medicine. 

As you will see, the reviewers were very positive about the paper and the importance of these follow-up data, but they raised a number of questions about specific study details and presentation. After discussing the paper with the editorial team and an academic editor with relevant expertise, I’m pleased to invite you to revise the paper in response to the reviewers’ comments. We plan to send the revised paper to all of the original reviewers, and of course we cannot provide any guarantees at this stage regarding publication.

When you upload your revision, please include a point-by-point response that addresses all of the reviewer and editorial points, indicating the changes made in the manuscript and either an excerpt of the revised text or the location (eg: page and line number) where each change can be found. Please submit a clean version of the paper as the main article file and a version with changes marked should as a marked-up manuscript. Please also check the guidelines for revised papers at http://journals.plos.org/plosmedicine/s/revising-your-manuscript for any that apply to your paper.

We ask that you submit your revision by the 5th of July. However, if this deadline is not feasible, please contact me by email, and we can discuss a suitable alternative.

Don’t hesitate to contact me directly with any questions (kjanin@plos.org). If you reply directly to this message, please be sure to ‘Reply All’ so your message comes directly to my inbox.

We look forward to receiving your revised manuscript. 

Sincerely,

Katrien Janin, PhD

PLOS Medicine

plosmedicine.org

1) We share the raised concern, specifically when it comes to the confounders included in your study. If indeed only the socioeconomic position of the parent is taken into account, as one of the reviewer writes, it would be “... difficult to be confident as to whether the claimed effects are due to ALD, or to (numerous) other confounders (e.g. self/family history of diseases, self alcohol/drug consumption, exercise, etc.) recognized to possibly have effects on the target outcomes (i.e. alcohol-related diagnoses, substance abuse, psychiatric diseases, etc.)” 

2) We also wondered about the use of the diagnosis date (given that the unhealthy alcohol use would almost certainly have preceded this.

3) Please avoid the use “alcohol-related liver disease offspring” in the manuscript and remove ‘nearly lifelong’ from the abstract.

More general request:

Data Availability 

PLOS Medicine requires that the de-identified data underlying the specific results in a published article be made available, without restrictions on access, in a public repository or as Supporting Information at the time of article publication, provided it is legal and ethical to do so.

Please see the policy at  

 http://journals.plos.org/plosmedicine/s/data-availability  

and FAQs at  

http://journals.plos.org/plosmedicine/s/data-availability#loc-faqs-for-data-policy   

PLOS defines the “minimal data set” to consist of the data set used to reach the conclusions drawn in the manuscript with related metadata and methods, and any additional data required to replicate the reported study findings in their entirety. Authors do not need to submit their entire data set, or the raw data collected during an investigation. Please submit the following data: 

 The Data Availability Statement (DAS) requires revision. For each data source used in your study:  

In your case, please include an email contact address on how one can make such a request from Statistics Denmark (www.dst.dk)

Reporting guidance 

Please report your study according to the relevant guidance which can be found here https://www.equator-network.org/reporting-guidelines/ 

Please ensure that the study is reported according to the STROBE guideline, and include the completed STROBE checklist as Supporting Information. Please add the following statement, or similar, to the Methods: ""This study is reported as per the Strengthening the Reporting of Observational Studies in Epidemiology (STROBE) guideline (S1 Checklist).

Prespecified analysis plan/study protocol 

Did your study have a prospective protocol or analysis plan? Please state this (either way) early in the Methods section.  

For all observational studies, in the manuscript text, please indicate: (1) the specific hypotheses you intended to test, (2) the analytical methods by which you planned to test them, (3) the analyses you actually performed, and (4) when reported analyses differ from those that were planned, transparent explanations for differences that affect the reliability of the study's results. If a reported analysis was performed based on an interesting but unanticipated pattern in the data, please be clear that the analysis was data-driven. 

Abstract layout 

Please structure your abstract using the PLOS Medicine headings (Background, Methods and Findings, Conclusions). In the last sentence of the Abstract Methods and Findings section, please describe the main limitation(s) of the study's methodology.

Author summary 

At this stage, we ask that you include a short, non-technical Author Summary of your research to make findings accessible to a wide audience that includes both scientists and non-scientists. The authors summary should consist of 2-3 succinct bullet points under each of the following headings: 

Why Was This Study Done? Authors should reflect on what was known about the topic before the research was published and why the research was needed.

What Did the Researchers Do and Find? Authors should briefly describe the study design that was used and the study’s major findings. Do include the headline numbers from the study, such as the sample size and key findings.  

What Do These Findings Mean? Authors should reflect on the new knowledge generated by the research and the implications for practice, research, policy, or public health. Authors should also consider how the interpretation of the study’s findings may be affected by the study limitations. In the final bullet point of ‘What Do These Findings Mean?’, please describe the main limitations of the study in non-technical language.

The Author Summary should immediately follow the Abstract in your revised manuscript. This text is subject to editorial change and should be distinct from the scientific abstract. Please see our author guidelines for more information: https://journals.plos.org/plosmedicine/s/revising-your-manuscript#loc-author-summary 

Introduction layout 

Please address past research and explain the need for and potential importance of your study. Indicate whether your study is novel and how you determined that. If there has been a systematic review of the evidence related to your study (or you have conducted one), please refer to and reference that review and indicate whether it supports the need for your study. 

Discussion layout 

Please present and organize the Discussion as follows: a short, clear summary of the article's findings; what the study adds to existing research and where and why the results may differ from previous research; strengths and limitations of the study; implications and next steps for research, clinical practice, and/or public policy; one-paragraph conclusion. 

Statistical reporting 

Please quantify the main results with 95% CIs and p values. 

When reporting p values please report as <0.001 and where higher as p=0.002, for example. When reporting 95% CIs please separate upper and lower bounds with commas instead of hyphens as the latter can be confused with reporting of negative values. Please include any important dependent variables that are adjusted for in the analyses. 

Supplementary materials  

Please note that supplementary materials are not checked and will be posted as supplied by the authors. Therefore, please double check. Please cite your Supporting Information as outlined here: https://journals.plos.org/plosmedicine/s/supporting-information - Please note you may use almost any description as the item name of your supporting information as long as it contains an "S" and number. For example, “S1 Appendix” and “S2 Appendix,” “S1 Table” and “S2 Table. Please ensure each supplementary material has a call out (link) from your main manuscript. 

Comments from the reviewers:

Reviewer #1: In this manuscript, the authors examine whether offspring of parents with alcohol-related liver disease had a higher incidence rate of hospital utilization across a series of reasons (alcohol-specific, non-alcohol abuse, injury, etc) using data from the National Denmark cohort registry data. The authors use a sample where offspring are matched to comparators based on age and sex. The results show that offspring of parents with alcohol-specific liver disease had higher incidence rate of contacts with hospitals across a range of reasons, and these associations appear to be stronger among those parents with lower educational attainment. Overall, the findings are interesting, but there are also several concerns and questions regarding the study that should be addressed. These are described below. 

(1) The sample included 60804 offspring. How many of them were siblings from the same nuclear family? Or was one offspring randomly chosen? Additional information on the sampling strategy is needed.

(2) The method states that age- and sex-matched comparators were selected and included in the analyses (20:1 design). It would be helpful to clarify the decision of matching only based on these two demographic factors. It was unclear why the comparators were not matched based on parental socioeconomic status (as indexed by educational attainment). 

(3) I believe that the socioeconomic status variable is operationalized as parent's educational attainment. This needs to be clarified throughout the manuscript. Additionally, it states that Table S1 provides a full description of the educational categorization, but this information appears to be missing. This information is crucial, especially in view of results surrounding differences by socioeconomic strata. 

(4) Some rationale and justification for using parents' educational attainment as the socioeconomic status. There is social mobility, both inter-generational and intra-generational mobility. Additional clarification would be helpful in understanding this analytic decision. 

(5) There appears to be some sex differences in the incidence rate ratios. It might be helpful to estimate the IRRs by sex. 

(6) The self-controlled case series analyses are interesting. For a general readership, a more detailed description of this method would be good. 

(7) How much comorbidity is there among the adverse health outcomes? Some sensitivity analyses censoring all cases with hospital contacts for multiple reasons might provide some robustness check. 

(8) The author note that "our findings shown in Figure 1 provide support for a proposed pathway for socioeconomic inequality in health: low socioeconomic position in childhood leads to poor mental health in adolescence; the poor mental health leads to heavy alcohol consumption and smoking in adulthood, and they lead to chronic diseases." While the associations appear to be stronger among those with lower parental SES, I think the results of this study does not necessarily provide direct support for this process-oriented hypothesis. I would revise the text and avoid drawing "causal" conclusions. 

Reviewer #2: "Adverse health outcomes in offspring of parents with alcohol-related liver disease: nationwide Danish cohort study" analyzes how a parent's diagnosis of alcohol-related liver disease (ALD), affects their offspring in terms of adverse health outcomes within the first year after said diagnosis. National Danish health registry data from 1996-2018 was employed, which provided data for over 60,000 affected offspring. Against age/sex-match comparators, it was found that parents' diagnosis did in fact correlate with an increase risk for adverse health outcomes, particularly when the offspring were of low socioeconomic position.

The findings are relevant towards providing appropriate preventative intervention for vulnerable offspring of ALD parents. However, some issues might be considered:

1. The study focuses on "the first year after the parent's ALD diagnosis". It might be commented on whether there might be a (considerable and varying) time lag between the incidence of ALD, and official diagnosis. This would be helpful in considering whether the impact on offspring is due to presence of ALD itself (in which case subsidiary analysis of health outcomes somewhat before official diagnosis might be warranted), or the official diagnosis.

2. Related to the above, analysis of incidence rate (and ratios and differences) against the offspring's previous self (in addition to matched comparators) before the ALD diagnosis, might be appropriate.

3. The significance of the ALD diagnosis might be explained further. Does it generally correlate to a significant change in the patient's ALD status (e.g. diagnosis implies incapacitation or other functional disability, since hospitalization is sometimes used as a synonym, such as in the Interpretation subsection of the Abstract)? This is as one might expect ALD to be a long-term ongoing risk occuring only after a prolonged period of alcohol (ab)use.

4. In the Introduction section, the hypothesis of whether risk to offspring after they have left home was stated as being less-investigated. It might thus be considered to perform stratified analyses on differential risk for offspring still living with parents, and those who do not.

5. In the Methods section, it was stated that comparators were matched to offspring on age, sex and birth year. The significance of including both age and birth year might be explained further, since this seems possibly largely redundant.

6. The full list of confounders might be explicitly specified. From the Data Source section and Table 1, the only confounder appears to be "socioeconomic position of parent (and not the offspring)", which moreover appears to be entirely education-based. Given this, it appears difficult to be confident as to whether the claimed effects are due to ALD, or to (numerous) other confounders (e.g. self/family history of diseases, self alcohol/drug consumption, exercise, etc.) recognized to possibly have effects on the target outcomes (i.e. alcohol-related diagnoses, substance abuse, psychiatric diseases, etc.)

7. In the Abstract, the sentence "Offspring of parents with ALD had a nearly lifelong higher rate of health outcomes..." might be rephrased for clarity. In particular, what does "lifelong higher rate" refer to here, since the incidence rate appears to apply only to the period after ALD diagnosis?

Reviewer #3: This appears to be a revised paper (indicated as PLOSMEDICINE-D-24-01249R1) but I did not review the original submission if this is the case. 

This is a concise paper. It uses a robust administrative data resource (Danish health registry data) to assess the risk of offspring of adults diagnosed with alcohol-related liver disease experiencing adverse health outcomes. It has a longitudinal design, with a case-comparator match design (ratio 20:1). The aim of the paper is to assess whether there is an increased risk for these offspring and when this might be greatest with a view to seizing the opportunity to intervene in order to reduce likelihood of experiencing these adverse outcomes in the future. 

There are a few study design points that are somewhat unexpected for a submission in a publication such as PLOS Medicine. For example, the pre-registration of study design and a priori hypotheses would be expected (although restrictions regarding the sharing of this study data are expected given the sensitive nature of the data and the need for researchers to apply directly to access the data for analysis in secure settings). The research questions are exploratory in nature, although this isn't explicitly stated. It might be advisable to include this detail.

Other study strengths include the use of ICD diagnoses for a range of study outcomes (adverse health events, not just alcohol-specific injury or diagnosis). These are listed clearly in the Supplementary document for interested readers. 

The self-controlled case series analyses appear to provide the most robust evidence of the 'unique' timing of the one-year interval post-parental diagnosis of alcohol-related liver disease as the highest risk period for offspring (especially those in mid-adulthood). That the risk of adverse outcomes for offspring of those with alcohol-related liver disease compared to other serious health conditions is elevated and persists over the lifespan is an interesting finding. It is also highlighted the compounding effect of low socioeconomic status has for individuals growing up/support by this type of family environment (where a parent has been a heavy/chronic drinker for many years results in liver disease) is not unsurprising but demonstrated empirically via this robust study design. 

I don't have any additional points to improve the quality of the current submission. In my view, it is a relatively simple exploratory paper. The analyses appear to have been well conducted and the take-home message is rightly tentative with respect to potential opportunity/window to intervene with offspring of adults who are diagnosed with alcohol-related liver disease. The fit with PLOS Medicine appears appropriate.

Minor comments:

Under 'Results' first paragraph - 1,216,07 match comparators should be 1,216,047 (p.8 of manuscript)

[LINK]

---

## [Decision Letter · Decision Letter 2]

4 Sep 2024

Dear Dr. Askgaard,

Thank you very much for re-submitting your manuscript "Adverse health outcomes in offspring of parents with alcohol-related liver disease: nationwide Danish cohort study" (PMEDICINE-D-24-01249R2) for review by PLOS Medicine.

Thank you for your detailed response to the editors' and reviewers' comments. I have discussed the paper with my colleagues and the academic editor, and it has also been seen again by the of the original reviewers. The changes made to the paper were mostly satisfactory to the reviewers. However, the editorial team concurs and some additional questions and requests. Therefore, we ask you to carefully address the comments in a further revision to preclude the need for further revisions and satisfy the editors and academic editor. When submitting your revised paper, please again include a detailed point-by-point response to the comments.

[LINK]

If you have any questions in the meantime, please contact me (kjanin@plos.org) or the journal staff on plosmedicine@plos.org.  

We look forward to receiving the revised manuscript by Sep 18 2024 11:59PM.   

Sincerely,

Katrien Janin, PhD

Senior Editor 

PLOS Medicine

plosmedicine.org

Comments from Academic Editor:

This is a large work and ALD is an important topic, likely to be the major cause of liver-related death in the coming decades as viral hepatitis control gets more accelerated.

However, my main issues with this work are as below:

1. What is the potential mechanism? It seems a lot are due to socioeconomic and social development factor. For example, the highest rates of problems in offspring occur in the first year from parents' diagnosis. This is likely due to parents being sick and thus diagnosed, absent from home, consuming family financial resources, all of which ultimately lead to resource (both financial and nonfinancial) deprivation which then lead to a variety of problems in offspring. It is hard to infer any biological association from this.

2. What is the clinical relevancy of this study conclusion? How is this going to change practice? I do not believe this study data will change practice. We already know that people with ALD have poor coping skills, poor medical adherence, and often lack social support, all of which make poor parenting and poor family/living situation for children or anyone living with them. It is also already well known that a positive family history of alcoholism is a risk factor for alcohol use disorder in offsprings, so offsprings of people with alcohol use disorder whether they have ALD or not should be offered additional social and school support to cope with their family environment.

(editorial add-on: point 2 should be more acknowledged)

Upon re-read the findings a few times: There were various statistical methods of matching and design performed in a large pt cohort. However, it seems to me the main results are higher risk of alcohol related hospital contacts for offsprings of parents with ALD, worse with lower socioecon and lower educational levels. It is well known that alcohol use disorder is associated with lower economic status and lower educational level. It is also well known that family history of alcohol use disorder is a risk factor for someone to develop AUD and consequently alcohol related health complications. The etiology of AUD is thought to involve both genetic and environmental factors (Kranzler, Am J Psy 2023, PMID: 37525595). 

In smaller detail, in response to editor comment #2 in Revision 2, the authors expanded analyses to include health outcomes 5 years before ALD diagnosis and 10 years after and found the results to be stable. This would suggest that that it was not that index ALD diagnosis analyzed in this study that was the issue but the likely lifelong exposure to alcohol use disorder and associated family dysfunction and other genetic factors that were the problems.

Comments from Reviewers:

Reviewer #1: My comments have been adequately addressed. I think this manuscript will make a fine contribution. 

Reviewer #2: We thank the authors for addressing our previous comments, and acknowledge our misconception for the self-controlled case series design originally raised in Point 2. Concerns on sociodemographic factors were covered in the responses to Reviewer #1.

Reviewer #3: I am happy the authors have addressed my requests for minor revisions.

The paper is suitable for publication now, in my opinion.

[LINK]

---

## [Editor Report · Decision Letter 3]

25 Sep 2024

Dear Dr. Askgaard,

Thank you for re-submitting your manuscript, "Adverse health outcomes in offspring of parents with alcohol-related liver disease: nationwide Danish cohort study" (PMEDICINE-D-24-01249R3) to PLOS Medicine and for your responses to the comments from the academic editor. 

I have discussed the paper with the academic editor, and I’m pleased to say that we plan to accept the paper for publication, pending some additional editorial questions and modifications, as outlined at the end of this email. We ask that you address these remaining points in one final revision. As before, please include a point-by-point response to these editorial requests and specify any changes made in the manuscript as a result. Please submit a clean version of the paper as the main article file, and a tracked version as a marked-up manuscript file.

Please also check the guidelines for revised papers at http://journals.plos.org/plosmedicine/s/revising-your-manuscript for any that apply to your paper. 

A reminder that when your manuscript is accepted, an uncorrected proof of your manuscript will be published online ahead of the final version, unless you've already opted out via the online submission form. If, for any reason, you do not want an earlier version of your manuscript published online or are unsure if you have already indicated as such, please let the journal staff know immediately at plosmedicine@plos.org.

We expect to receive your revised manuscript within 1 week (Oct 2nd). Please email me directly at hvanepps@plos.org if you have any questions or concerns.

Kind regards,

Heather

Heather Van Epps, PhD

Executive Editor 

PLOS Medicine

hvanepps@plos.org

Requests from Editors:

1. Please add line numbers to the manuscript and update the STROBE checklist to specify line numbers for each item.

2. Financial disclosure statement: please consider changing “donated to” to “awarded to”

3. Data Availability Statement: you have indicated that the code used in the analyses will be available upon reasonable request. We strongly encourage all authors to share code supporting the findings in their manuscript. In cases where the data are unavailable, we recommend also sharing a synthetic dataset that can be used to check the veracity of the code. I also noticed that you accessed the data via a Secure Research Environment and this may limit the code files that you could download and share. If this is the case and your ability to share code is limited by an access agreement, then please state this in the Data Availability Statement. If you are unsure what can be shared or how to share code we are happy to support you.

4. Author summary, “What did the researchers do and find” section. Please consider modifying the final bullet point for clarity as follows: “In the year after the parent’s diagnosis of alcohol-related liver disease, offspring experienced an increase in hospitalization due to outcomes associated with self-harm (e.g. poisoning) and poor mental health (e.g. psychiatric disease).” (or similar).

5. Results section: Throughout this section, please indicate absolute number wherever percentages are reported [N (%)]. 

6. Table 3: please consider removing the column heading given that it is redundant with the title of the table.

7. Table 3: the note that is included below the table would be better placed as a Discussion point; please remove this from the table.

---

## [Editor Report · Decision Letter 4]

3 Oct 2024

Dear Dr Askgaard, 

On behalf of my colleagues and the Academic Editor, Mindie Nguyen, I am pleased to inform you that we have agreed to publish your manuscript "Adverse health outcomes in offspring of parents with alcohol-related liver disease: nationwide Danish cohort study" (PMEDICINE-D-24-01249R4) in PLOS Medicine.

PRESS

Kind regards,

Heather 

Heather Van Epps, PhD 

Executive Editor 

PLOS Medicine